# HIV Latency and Nanomedicine Strategies for Anti-HIV Treatment and Eradication

**DOI:** 10.3390/biomedicines11020617

**Published:** 2023-02-18

**Authors:** Mickensone Andre, Madhavan Nair, Andrea D. Raymond

**Affiliations:** 1Department of Immunology and Nanomedicine, Herbert Wertheim College of Medicine, Florida International University, Miami, FL 33199, USA; 2Institute of Neuroimmune Pharmacology, Herbert Wertheim College of Medicine, Florida International University, Miami, FL 33199, USA

**Keywords:** HIV-1 latency, ARVs, nanomedicine, exosomes, shock and kill, block and lock

## Abstract

Antiretrovirals (ARVs) reduce Human Immunodeficiency Virus (HIV) loads to undetectable levels in infected patients. However, HIV can persist throughout the body in cellular reservoirs partly due to the inability of some ARVs to cross anatomical barriers and the capacity of HIV-1 to establish latent infection in resting CD4+ T cells and monocytes/macrophages. A cure for HIV is not likely unless latency is addressed and delivery of ARVs to cellular reservoir sites is improved. Nanomedicine has been used in ARV formulations to improve delivery and efficacy. More specifically, researchers are exploring the benefit of using nanoparticles to improve ARVs and nanomedicine in HIV eradication strategies such as shock and kill, block and lock, and others. This review will focus on mechanisms of HIV-1 latency and nanomedicine-based approaches to treat HIV.

## 1. Introduction

Human Immunodeficiency Virus (HIV) disease progression to acquired immunodeficiency syndrome (AIDS) is impeded by antiretroviral drugs (ARVs) that inhibit critical steps in the HIV life cycle [1]. However, ARVs do not target or eliminate the viral genome integrated into the cellular genome. Once integration occurs, curing people becomes extremely difficult because a small percentage of HIV-infected cells remain dormant (latent) or suppressed at specific anatomical sites throughout the body. Within these reservoirs, HIV-infected cells can be suppressed by ARVs, persistent or latent. HIV-persistent cells produce few virions, and cells latently infected do not produce new virions but can produce viral transcripts and proteins or remain transcriptionally silent [2]. However, once reactivated or upon ARV treatment interruption, these cells can rebound HIV viral load to detectable levels within two weeks [3]. 

Resting memory CD4+ T-cell is one of the most common latent infected cells that can be further separated into different subsets [4]. HIV latency usually occurs in CD4+ T-cells transitioning from their effector state to their long-lived resting memory state [5]. During this transition, HIV can infect a cell and integrate its viral DNA into the host genome. The integration process is random, and multiple integrations can occur in one cell. Integration can occur while the host transcription factors (AP-1, Sp1, and NF-κB) become less available. As a result, the provirus will become latent in these long-lived memory CD4+ T-cells that usually have a half-life of about 44 months [6,7]. Other cell types can also harbor the virus, such as macrophages and monocytes. Many of these HIV cellular reservoirs are located throughout the body, but identifying and measuring HIV reservoirs is challenging with current technologies. 

Nanomedicine is a novel and revolutionary strategy to treat complex diseases. Nanomedicine offers many advantages over conventional medicine, such as being more biocompatible, capable of delivering drugs to specific pathological tissues and possessing a high drug-loading capacity [8]. In addition, nanoparticles can also increase drug stability and control the release of the drug. With all of these properties, transitioning nanomedicine to a clinical setting has been hampered. One of the major challenges of nanomedicine transition to a clinical setting happens occurs when positive preclinical results do not meet expectations in clinical settings [9]. Nonetheless, some nanoparticles, such as liposomes, are becoming rapidly available to the public, such as with the 2020 breakthrough COVID mRNA vaccine [10]. Currently, researchers are exploring utilizing nanoparticles to treat HIV infection. This review will discuss mechanisms of HIV latency and the latest nanomedical strategies to treat and eliminate HIV infection. 

## 2. Molecular Mechanisms of HIV Latency

At the beginning of HIV cellular infection, HIV glycoprotein (gp120) binds to the host CD4 receptor. HIV can then enter the cell in two common pathways. In the primary pathway, gp120 undergoes a conformational change and exposes the gp120’s V3 domain. The V3 domain binds to the host chemokine coreceptors, such as CCR5 or CXCR4, depending on the viral tropism caused by a point mutation [11]. A post-conformational change dissociates gp120 from gp41, which facilitates fusion with the host membrane [12]. The second pathway involves endocytosis. Upon gp120 binding to the CD4+ receptor, the virus enters the cell via an endocytic vesicle [13]. Chauhal et al. demonstrated that HIV-1 enters astrocytes via the endocytosis pathway in which gp41 facilitates fusion with the luminal membrane of the endosome. The endocytosis pathway has been reported to produce more defective viruses [14,15,16]. Once fusion occurs in both pathways, the capsid is released into the cytosol.

Once in the cytosol, the capsid advances to the nucleus via microtubules [17]. How the HIV genome dissociates from the capsid is poorly understood. The timing and location of uncoating are debated among researchers. The old dogma considered uncoating to occur in the cytosol; conversely, new evidence demonstrated that uncoating could occur inside the nucleus near the integration site [18,19,20,21]. In agreement with this notion, in 2020, researchers used a novel technique, HIV-1 ANCHOR, to demonstrate the reshaping of HIV’s capsid proteins into a pre-integration complex (PIC) to enter through the nucleus pore [22]. It is also believed that reverse transcriptase, located inside the capsid, finishes transcribing the viral RNA into DNA inside the nucleus [18,19,20]. After transcription, the new viral DNA is incorporated into the host DNA via integrase. This integrated viral DNA is called HIV proviral DNA/provirus. At this stage, integration can induce cellular apoptosis or promote/suppress proviral transcription [11,23]. Trans-Activator of Transcription (Tat) and Rev are essential for proviral transcription and nuclear export of viral mRNAs, respectively. Rev alone can also upregulate the expression of HIV structural proteins and downregulates the expression of the Tat protein [24].

Cells latently infected with HIV establish reservoirs at specific anatomical sites such as the brain [25]. At these reservoirs, two types of latency exist, pre-integration and post-integration (Figure 1). Pre-integration latency primarily occurs when the host cells transition to a quiescent state before HIV viral DNA is integrated. The mechanisms of pre-integration latency include poor nuclear transportation of the PIC, defective viral proteins, and the state of the cell life cycle [22]. In addition, the viral PIC can remain stable for several weeks on centrosomes and possibly integrate into the host genome when the host cell becomes reactivated [26]. Pre-integration latency is less stable than post-integration latency. Post-integration latency occurs after the viral genome has integrated into the host genome and remains transcriptionally silenced [27]. Memory CD4+ T-cells transitioning from an active to a quiescent state also exhibit post-integration latency. Several factors can contribute to this state, such as low transcriptional factors, DNA modification, transcriptional suppression, and low Tat and P-TEFb (Figure 1). 

Post-integration latency can depend on the expression threshold of the HIV Tat protein, which forms a positive feedback loop that upregulates HIV transcription. When Tat expression is above the threshold, transcription can occur. However, when Tat expression is below the threshold, the provirus is considered dormant/latent. As a result, HIV LTR transcription is significantly reduced/inhibited. Several factors can contribute to the low expression of Tat, such as low levels of host transcriptional factors, the presence of host transcriptional repressors, and the integration of the HIV genome into a heterochromatin region [28]. For example, in microglia, Sp3 (a host transcriptional repressor) is regulated and induces latency by blocking HIV transcription. However, latency in astrocytes is induced by low expression of Sam68 or by high expression of cellular proteins with Rev-interacting domain (Risp), class I histone deacetylases (HDACs), and a lysine-specific histone methyltransferase, Su(var)3–9 [29,30,31]. Additionally, the microenvironment of the cell can also determine latency. Zhuang et al. observed that under hypoxic conditions, hypoxia-inducible factors could restrict HIV transcription by binding to the viral promoter regions [32]. 

There are multiple types of HIV infections including latent, active, and persistent (Figure 2). Persistent infection is believed to result from low ARV concentrations in anatomical HIV reservoirs. Within these reservoirs, HIV-infected cells can have transcriptionally active proviruses that are persistent (producing little virions) or latent (producing no virions) [33]. Moreover, infected cells can contain replication-competent or defective proviruses [34]. Replication-competent proviruses produce virions with the ability to replicate. On the other hand, defective proviruses produce imperfect proteins that prevent the virions from reproducing. Approximately 5–11% of CD4 T-cells within the reservoirs are replication-competent [35]. 

These reservoirs are established throughout the body where ARVs are limited [36] (Figure 3). In some unique microenvironments such as the CNS, compartmentalization can occur, resulting in HIV variants. In addition, infected cells can also undergo normal cellular division (clonal expansion), which replication-competent proviruses can increase in number [37]. It is estimated that more than half of the latent cells are maintained by clonal expansion [38,39]. Antigen-driven proliferation is considered one of the forces that drive clonal expansion. The location of these replication-competent cells needs to be identified to develop an effective strategy to prevent viral rebound. Specific tissues within the body serve as reservoirs and possess persistent and latently infected cells, described below. 

### 2.1. Gut-Associated Lymphoid Tissue (GALT)

The GALT is believed to be the first major organ infected by HIV and contains the most latently infected cells [40]. Most of the body’s lymphocytes are in the intestine [41,42]. In a study, Yukl et al. found that HIV RNA and DNA were 5–10-fold higher in the GALT than in the peripheral blood [43]. Moreover, in 2017, Estes et al. demonstrated in primates that the GALT harbors the majority of SIV infection, with 62.3% of SIV reservoirs before therapy and 98% after [44]. Another group of researchers developed an immunoPET to identify HIV infection. The gut and the lymph nodes were the sites with significantly latent cells [45]. The HIV cellular reservoir in the GALT has been identified as CD4+ T-cells, macrophages, B cells, and dendritic cells (DC) [46,47,48,49].

### 2.2. Lymph Nodes (LNs)

The lymph nodes (LNs) are considered the second-highest site with latently infected cells. This notion suggests that the LNs should be a major site to target and eradicate latently infected cells that mainly contribute to viral rebound. Estes et al. supported this notion by demonstrating that the LNs containing 35.9% of SIV latently infected cells before therapy and 0.53% after [44]. Although with a small percentage and deemed the second site with latently infected cells, the LNs had demonstrated to contribute most of the viral rebound during ARV treatment interruption. In the study, the GALT and blood were the second and third contributors to the viral rebound [50]. The findings in primates also suggest that the LNs contain most of the replication-competent proviruses while the GALT includes mostly detective proviruses. In another study, PMBC contained a similar level of HIV DNA copies to the ileum. However, when examining the ratio of HIV RNA to DNA between the blood and gut, the PBMC level was significantly higher. This demonstrated that more HIV virions were released from the blood [51]. 

### 2.3. Central Nervous System (CNS)

From the periphery, HIV-infected CD14+ and CD16+ monocytes are the primary cells to cross the blood–brain barrier (BBB) and transmit HIV infection into the CNS [29,52,53]. Nevertheless, CD4+ T-cells were also reported introducing HIV-1 into microglial cells [53,54]. Furthermore, in a study by Albalawi et al., a novel subgroup of T-cells expressing CD4 and CD8 had the capability to migrate into the brain [55]. 

Microglia are considered the primary latently infected cells in the CNS because of their long half-life and capability of producing replication-competent virions. Perivascular macrophages can also produce HIV virions; however, these cells are not considered a main reservoir because they have a short half-life of months. Cochrane et al. reported that CD68+ myeloid cells harbor HIV DNA. The group examined the autopsy of the frontal lobe of viremic or virally suppressed individuals. There was no difference in the level of total HIV DNA between the viremic group and the virally suppressed group [56]. The pericytes are also susceptible to HIV infection, and the virus can replicate in pericytes [57,58,59]. Although astrocytes are the most numerous cells in the CNS, it is debatable whether HIV can infect astrocytes and replicate [59]. Few studies have shown that astrocytes can produce new virions, while other studies have found that HIV does not infect astrocytes based on no evidence of HIV DNA in astrocytes [59]. Conversely, with improved detection and quantification methods, integrated HIV DNA was found in a small population of astrocytes. The researcher reported that astrocytes had poor infectivity but were extremely effective at cell-to-cell transmission of HIV [60]. Some studies indicate that astrocytes are persistently infected and can reseed HIV into peripheral organs [61,62,63]. Su et al. took brain cells latently infected with HIV from mice on ARVs and intraperitoneally injected them into uninfected mice. After eight weeks, HIV-1 RNA levels were detected in the mice, suggesting that a replication-competent provirus can be established in astrocytes despite ARV treatment [64]. 

Moreover, infected glial cells can release exosomes, an extracellular vesicle containing HIV viral proteins such as the Nef [65,66]. These exosomes can negatively impact neighboring cells, fundamentally contributing to HIV-associated neurocognitive disorder (HAND) [65]. Egger et al. reported that 50% of HIV+ patients taking ARV suffer from HAND [67], suggesting that HIV-infected glia can indirectly contribute to HIV neuropathology.

### 2.4. Other Tissue Reservoirs

Although containing a small percentage of HIV reservoirs (the spleen, kidney, bone marrow, blood, and genital tract (Figure 3)), other tissues need to be considered for therapeutic approaches because these reservoirs can potentially reseed or reinfect other tissue [64]. In fact, it is believed that CD11c+ epidermal dendritic from the genital tract is the first cell to get infected and transfer the infection throughout the body [68]. In addition, infected cells from the endometrium have been found to transcribe more HIV than the gut, cervix, and liver reservoirs, suggesting that this reservoir can increase the viral load [51]. Additionally, these reservoirs can create new variants that contribute to compartmentalization, as observed in the male genital tract of all the participants in one study [69].Some of the other tissue reservoirs include the renal epithelium (kidneys) and mast cells (lungs). Viral replication during ARV treatment was observed in these cells [70,71,72,73]. In addition, new biomarkers are also being discovered in these reservoirs to target latently infected cells. Recently, a study suggested that HIV-infected cells in the blood can be targeted via the CD32a receptor, found on 0.012% of infected CD4+ T-cells. Although it is a small population of CD4+ T-cells, these CD32a+ cells make up most of the replication-competent proviruses [74]. CD127 is another biomarker highly expressed on cells with replication-competent proviruses, suggesting it is another biomarker of T-cell latency [75]. 

## 3. Anti-HIV Therapy

### 3.1. Antiretroviral Therapy (ART)

Current anti-HIV therapies focus on inhibiting essential steps in the HIV life cycle; nonetheless, HIV can mutate, rendering these drugs useless if taken alone. HIV treatment is usually given in combination with two or three groups of ARVs, called cART. There are five classes of ARVs labeled as non-nucleoside reverse transcriptase inhibitors, protease inhibitors, entry/fusion inhibitors, integrase inhibitors, and nucleoside/nucleotide reverse transcriptase inhibitors. The three drugs of choice are an integrase strand transfer inhibitor and two nucleoside reverse transcriptase inhibitors [76]. ARVs are administrated daily, which can make adherence difficult. Any interruption of this daily regimen can result in the virus rebound. 

ARVs are administrated orally, making absorption the main route. Long-acting injectables (LAIs) such as Cabenuva on the other hand are administered via intramuscular injection. A higher concentration of the drug enters the systemic circulation via intramuscular injection than the oral route which gives LAIs and advantage over orally administrated drugs.

The biodistribution of the ARVs was also assessed. Labarthe et al. reported that in mice ARVs (tenofovir, emtricitabine, and dolutegravir) had the highest concentration in the digestive tract, liver, and kidneys but the lowest concentration in the brain [77]. Although the brain is deemed to have a low ARV concentration compared to other organs, a recent study that was the first to assess ARV concentration from human brain tissues reported a higher concentration than any published concentration [78]. Furthermore, different ARVs can be more concentrated in different tissues, indicating that certain steps in the HIV life cycle are not inhibited at certain reservoirs. For example, Rosen et al. also witnessed heterogeneous ARV disposition in lymph nodes [79]. 

Most ARVs are metabolized in the liver after absorption via the cytochrome P450 enzymes. Therefore, ARV treatment consider the drug-to-drug interactions, which might increase drug toxicity. Additionally, some people living with HIV utilize marijuana, medically or recreationally, which can inhibit the cytochrome P450 enzymes [80,81,82]. Ultimately this can lead to an increase in ARV concentration in the bloodstream increasing side effects and excretion rates. ARVs have a half-life in the body estimated to be between 2–9 h, so it is necessary to administer the drug daily to maintain a steady state. Most ARVs are eliminated via the kidney.

### 3.2. A Hematopoietic Stem-Cell Transplantation

After discontinuing ART in one year, a London and a Berlin patient were believed to be cured of HIV when clinicians observed no viral rebound [83,84,85]. Both patients received chemotherapy for cancer treatment, followed by hematopoietic stem-cell transplantation with cells containing the 32 base-pair in the CCR5(Δ-32 bp)deletion gene. Two other HIV-infected patients in Boston received a similar treatment but did not receive implanted cells without the Δ-32 bp deletion in CCR5, resulting in a viral rebound after only 3 and 8 months [85]. A mathematical model later demonstrated that these patient reservoirs had a two-log reduction compared to the Berlin patient’s 3.5-log reduction. This strategy shows that eradicating the viral reservoir is a possible route for a HIV cure; however, the method may be costly and impractical.

### 3.3. Shock and Kill/Kick and Kill

The shock and kill strategy functions to reactivate and eradicate latently infected cells. Latency reversal agents (LRAs) are drugs that reactivate latent cells. Consequently, these latent cells can be eliminated via virus-mediated cytolysis or immune-mediated clearance [86]. Different LRAs can activate transcription, modify chromatins, or facilitate transcriptional elongation. In 2020, N-803, an interleukin-15 super agonist, activated latently infected cells successfully in mice and primate models; however, due to the level of toxicity, the study was deemed unsafe for clinical trials [87]. In the RIVER study, a randomized clinical trial found no difference in replicated-competent provirus between HIV-positive patients on ART or their shock and kill treatment [88]. The study used vorinostat (shock) and a viral vector vaccine (kill) to target latent cells; however, the clinical trial did not provide evidence for latency reversal or ART interruption to assess viral rebound [89]. Overall, the shock and kill strategy lacks specificity. Current LRAs tend to cause global activation of both uninfected and infected T-cells. Moreover, eliminating the infected cells once activated is difficult. Improvements to the shock and kill strategy are needed for this form of therapy to be effective.

### 3.4. Block and Lock

The block and lock strategy aims to silence HIV provirus permanently via many mechanisms permanently; nonetheless, most techniques have only temporarily silenced HIV transcription. These mechanisms occurred by several molecules such as didehydro-cortistatin A, LEDGINs curaxin CBL0100, HSP90 Inhibitor, Jak-STAT Inhibitors, and ZL058 Tat inhibitor, all of which were shown to delay viral rebound [90]. These techniques are not specific to HIV proviral DNA and might alter other cellular functions. Therefore, researchers have used small interfering RNA (siRNA) to target regions in HIV LTR to epigenetically silence HIV transcription via histone deacetylation [91]. Overall, it is questionable whether the block and lock strategy can have specificity to lock HIV provirus permanently [90]. 

### 3.5. Anti-HIV Vaccines

The development of broadly neutralizing vaccines against HIV have been been disappointing. For the past 30 years, HIV vaccines have shown low efficacy and were strain-specific [92]. In 2014, the HIV Vaccine Trials Network (HVTN) outlined large-scale clinical trial directions to improve vaccine efficacy, but no successful vaccine has been developed. Moreover, it is questionable how vaccines will address HIV latency. In one study, a vaccine was paired with an LRA, resulting in 33% of the infected primates being able to control their infection virologically [93]. In 2023, the Phase 3 Mosaico HIV vaccine clinical trial with 3900 volunteers of men who have sex with men, the vaccine was deemed safe but ineffective. The vaccine was composed of multiple HIV subtypes. The vaccine candidate used the common-cold virus adenovirus serotype 26 for delivery. The study resulted in a similar infection rate between the placebo and vaccine groups. Since the success of the anti- SARS-CoV2 mRNA vaccine, researchers have shifted the HIV vaccine strategy to focus on the mRNA vaccine technology. This strategy uses lipid nanovesicles to transport mRNA which code for specific viral proteins. This may be a promising strategy for an HIV vaccine.

## 4. Nanomedicine in HIV Therapeutics

Nanomedicine has grown considerably during the last two decades. Nanoparticles typically range from l to 100 nanometers. In addition, the composition of nanoparticles can be inorganic or organic. Nanoparticles can have a variety of shapes to meet the experiment’s needs. Moreover, each type of nanoparticle offers a unique property with extraordinary functionalities. Some nanoparticle applications can range from biosensing to imaging and drug delivery (Figure 4). We have summarized HIV nanotherapeutics in Table 1. Nanoparticles’ unique functionalities are being exploited to treat HIV, which are mentioned below: 

### 4.1. ARV Drug Delivery

A wide variety of nanoparticles have been used for drug delivery in HIV studies. Nanoparticles can be conjugated with drugs and ligands to specifically target the cells expressing the receptors for the ligands and deliver the drugs. Ultimately, the drug will be more concentrated in these targets than if the drug was administered alone. Freeling et al. engineered liposomes, a lipid nanovesicle containing ARVs (anti-HIV LNP), composed of lopinavir, ritonavir, and a small percentage of tenofovir, to target CD4 T-cells in the LNs. The primates were subcutaneously dosed and showed elevated lopinavir and ritonavir for over seven days. This strategy resulted in a 50-fold higher intracellular tenofovir, ritonavir, and lopinavir drug in lymph nodes compared to the free drugs [101]. Poly(lactic-co-glycolic) acid (PLGA) nanoparticles can also encapsulate drugs. PLGA conjugated to α4β7 monoclonal antibodies and loaded with tipranavir (a nonpeptidic protease inhibitor) targeted gut-homing T-cells found in the GALT to block HIV infection. The nanocarriers were shown to reach the small intestines within six hours and were estimated to contain 40% of the tipranavir. Twelve hours post-administration, the nanocarrier had reached the α4β7+ lymphocytes found in the lamina propria [106]. Chitosan nanoparticles were used to deliver Dolutegravir, an integrase inhibitor. The nanoparticle improved drug solubility and concentration at crucial reservoirs [94]. Another type of nanoparticle called magnetoelectric nanoparticle (MENP) is a potent core-shell nanoparticle that can create an electric impulse in a magnetic field. MENPs were used to release an ARV drug in the brain externally and on-demand after crossing the BBB model without producing much heat [105].

In early 2021, the FDA approved the first monthly injectable long-acting cART called Cabenuva, composed of cabotegravir and rilpivirine [106,107]. Cabenuva is grouped in a class of drugs called long-acting injectables (LAI), which can be deemed as nanoparticles [59,60,61]. In a Phase 3 randomized trial, patients on Cabenuva had HIV-1 RNA levels below 50 copies/mL, similar to patients who took the daily oral cART therapy of dolutegravir–abacavir–lamivudine [108]. With this new treatment, patients’ adherence to their regimen should improve with Cabenuva since the treatment plan has been reduced to 12 days instead of 365 days with cART. Cabenuva should also lower the cost of medical treatment; HIV medicine is expensive, and cART can be a financial burden to low-income patients. In 2018, USA’s cART was ranked the fifth highest medical treatment, amounting to USD 22.5 billion in spending, ranging from USD 36,000 to USD 48,000 annually per person [109]. The high cost and the side effects from ARVs, such as nausea, diarrhea, headache, and chronic pain, greatly impact patients’ adherence to treatment [110]. However, ARV treatment is improving. 

Garrido and colleagues used gold nanoparticles (GNPs) conjugated to Raltegravir (an HIV integrase inhibitor) and demonstrated antiretroviral activity in treated cells. The GNPs were also able to cross the BBB and showed no toxicity [108]. In another study, Cabotegravir was loaded onto gold nanoparticles with an encapsulation efficiency of 97.2 ± 3.9%. Cabotegravir has a low bioavailability due to its poor absorption in gastric fluid. The drug release from the nanoparticle was tested in a phosphate buffer saline (PBS) and simulated gastric buffer. After nine hours, 45.5 ± 2.8% of Cabotegravir was released in PBS, and within 24 h, about 45.5 ± 2.8% was released in a gastric buffer. The Cabotegravir-gold nanoparticles were less toxic to HeLa and HEK 293 [99]. The low bioavailability of Tenofovir is attributed to its negative charge. Abadi et al. utilizing gold nanoparticles conjugated with Tenofovir resulted in low cytotoxicity, great biodistribution, and high anti-HIV activity [99]. Overall, the utilization of nanoparticles in anti-HIV nanomedicine has excellent potential for targeted delivery and low-toxicity treatment. 

### 4.2. Vaccines

Virus-like particles (VLPs), which mimic HIV’s multiple strains’ structure in its natural environment, have been used for vaccine treatment. VLPs are noninfectious and ongoing research in preclinical studies shows these nanoparticles are antigenic and can induce a potent immune recall response [111]. GNP is also used as a vaccine delivery system [112]. GNPs conjugated with Gag and p17 and high mannoside-type oligosaccharides enhanced dendritic cells (DC) antigen presentation. DC cells are important vaccine candidates because DC presents antigens to the adaptive immune system. A bio-nanocapsule composed of a hepatitis B virus envelope L protein conjugated with anti-CD11c was used to target Cd11c+ DCs. The researchers demonstrate humoral and cellular immunities without adjuvant [113]. Other types of vaccines, such as the mRNA or protein liposomes-based nanoparticle vaccine, are currently being evaluated for the potential to stop HIV.

### 4.3. Shock and Kill/Kick and Kill for HIV Eradication

Exosomes have been used in many clinical trials because they are non-toxic and biocompatible [114]. These organic lipid bilayer vesicles are released from most organisms and cell types [115,116,117]. Exosomes containing Tat have been shown to reactivate latently infected cells [118,119,120]. Iron oxide nanoparticles have been used to reactivate latently infected cells. Given the magnetic properties of iron oxide nanoparticles, the conjugation of ligands and drugs to iron oxide nanoparticles allows for site-specific tissue delivery via a magnet. Jayant et al. utilized iron oxide nanoparticles to magnetically guide an LRA and an ARV across an in vitro transwell BBB to target astrocytes. The p24 levels were reduced by 33% while cell viability remained over 90% five days post treatment demonstrating that using nanoparticles can improve the shock and kill treatment outcomes [121].

## 5. Conclusions and Perspectives

Overall, this review focused on mechanisms of HIV-1 latency and nanomedicine-based approaches to treat HIV and eliminate cells latently infected with HIV. The path to curing HIV infection is locating and eliminating the latent reservoirs that contain replication-competent proviruses. Despite reducing peripheral viral loads to undetectable levels, current ARV therapy has several limitations—low CNS penetration, high toxicity, high cost, and side effects. However, nanomedicine-based approaches can improve HIV therapy by enhancing drug delivery, inducing effective immune responses against HIV-infected cells, and most importantly directly targeting HIV latent reservoirs. Achieving a potential cure for HIV may be possible once treatments are paired with nanomedicine. Although nanomedicine has revolutionary potential to improve anti-HIV medicine, translating these nanomedicine-based therapeutics to the clinical setting has been slow. More studies to develop and better characterize the biodistribution, efficacy, and pharmacokinetics of nanotherapeutics are needed to expedite clinical translation and move the future of anti-HIV nanomedicine forward.

## Figures and Tables

**Figure 1 biomedicines-11-00617-f001:**
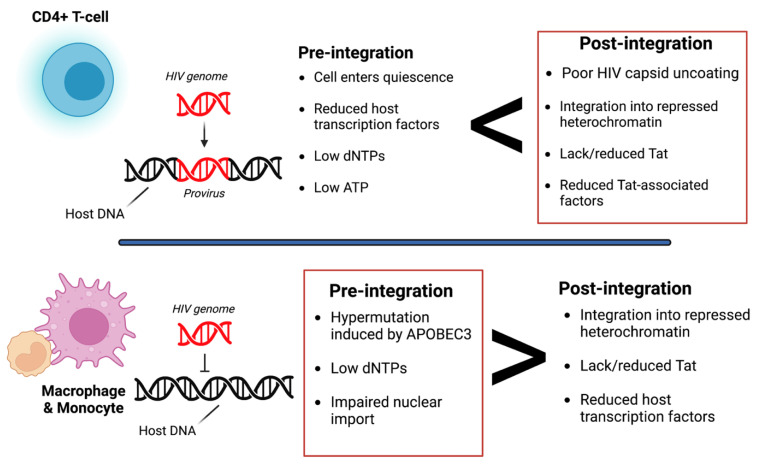
Overview of pre- and post-integration HIV latency. Pre-integration latency is more common in monocytes/macrophages due to poor nuclear transport of the pre-integration complex (PIC), change in cell cycle phase, and defective reverse transcriptase. Post-integration latency in memory T-cells during ARV treatment is partly due to transcriptional interference, low cellular transcription factors, DNA methylation, chromatin organization, and reduced p-TEFb and Tat protein. Red boxes indicate which type of latency is more common. It is also important to note that pre-integration is more common in the T-cells of untreated individuals. Image created with biorender.com and Adobe Illustrator/Photoshop.

**Figure 2 biomedicines-11-00617-f002:**
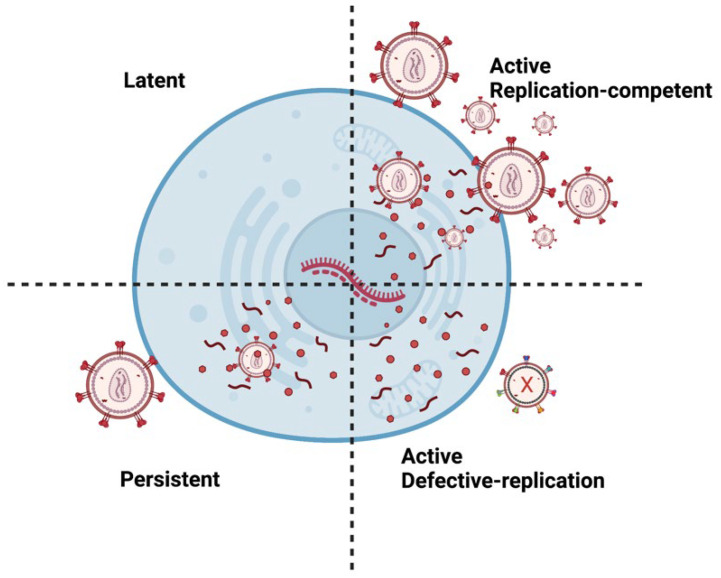
Different types of HIV Infection. A latent provirus is transcriptionally silent and produces no virions. However, active provirus can produce virions. Replication-competent provirus produces virions that can replicate. Persistent proviruses produce a few virions. Defective proviruses contain many mutations that produce nonfunctional proteins that render the virions incapable of replicating. It is important to note that latent or persistent provirus can be replication-competent or defective. Image created with biorender.com and Adobe Illustrator/Photoshop.

**Figure 3 biomedicines-11-00617-f003:**
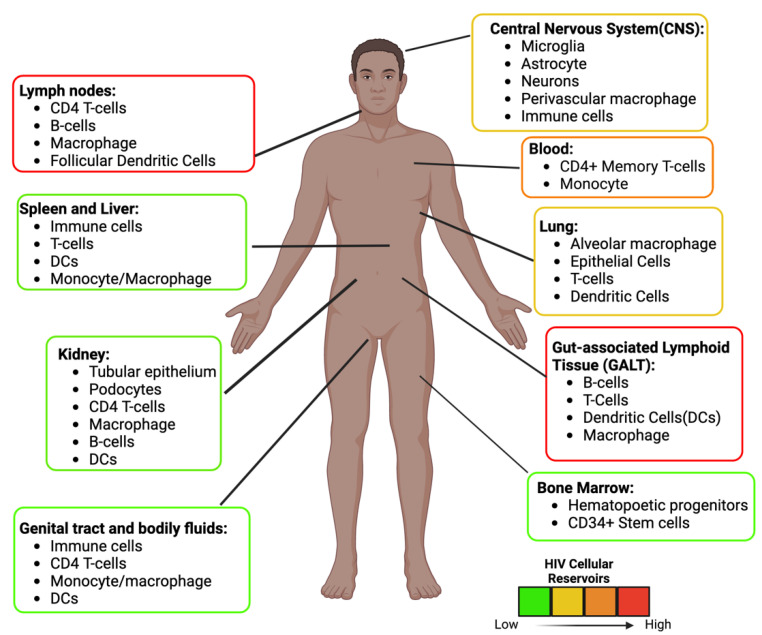
HIV Anatomical and Cellular Reservoirs during ART. Estimation of the location and type of cells latently infected with HIV. The color depicts the high (red) to the low (green) concentration of latent cells. The GALT and the Lymph nodes are two of the major sites. The estimation and cellular reservoirs were obtained from multiple sources. Image created with biorender.com and Adobe Illustrator/Photoshop.

**Figure 4 biomedicines-11-00617-f004:**
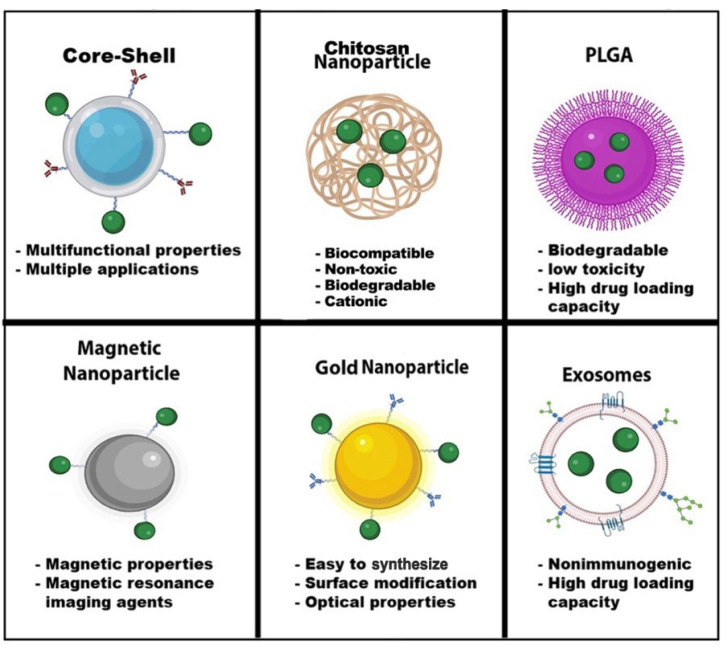
Nanoparticle Advantages. Nanoparticles have a wide range of advantages and applications in the field of nanomedicine. The shape, size, and composition give these nanoparticles unique properties for biosensors, drug delivery, MRI, diagnostic and therapeutics, optical imaging, and energy transfer. Some nanoparticles can have multiple functions, such as core-shell nanoparticles. Green dots—drugs. Image created with biorender.com and Adobe Illustrator/Photoshop.

**Table 1 biomedicines-11-00617-t001:** Summary of Nanoparticle type used in therapeutics against HIV infection and latency.

Nanoparticle Type	Drug/Agent	Target	Results	Reference
**Chitosan**				
	Dolutegravir Tenofovir alafenamide	HIV-infected cells	Dolutegravir became more soluble with the nanoparticle and had a higher concentration in multiple organs than drug alone. Extended release (56 %) of the drug for 16 days	[94] [95]
**Exosomes**				
	ARVs (Emtricitabine) Zinc Finger Protein (mRNA)	TZM-bl cells HC69.5	Reduced HIV infection Repressor-loaded anti-HIV-1 exosomes suppress virus expression	[96] [97]
**Gold**				
	Tenofovir Cabotegravir Raltegravir	TZM-bl cells HEK293 HeLa	~15-fold higher anti-HIV-1 reverse transcriptase activity than drug alone and great biodistribution Less cytotoxicity than the drug alone Across the BBB and displayed antiretroviral activity and no toxicity	[98] [99] [100]
**Iron oxide**				
	LRA	Astrocytes	Across the BBB, and 33% reduction in p24 level and cell viability over 90% after five days	[100]
**Liposomes**				
	ARVs (intracellular tenofovir, ritonavir, and lopinavir) HIV-1 Envelope	CD4 T-cells Env-specific B cells	50-fold higher intracellular ARV Efficiently activated Env-specific B cells	[101] [102]
**Magnetoelectric (MENPs)**				
	ARV drug	BBB model	Successfully crossed the BBB model and released the drug without producing heat	[103]
**Poly(lactic-co-glycolic) acid (PLGA)**				
	Protease inhibitor	Gut-homing T-cells	Successfully penetrated the reservoirs in the GALT more effectively than the free drug	[104]
**Virus-Mimicking Polymer**				
	ARV drug	CD169+ Macrophages	Achieved inhibition of HIV-1 infection of primary human macrophages for up to 35 days.	[105]

## Data Availability

Not applicable.

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
