# Peer review of "HIV Latency and Nanomedicine Strategies for Anti-HIV Treatment and Eradication"

_biomedicines, 2023, doi:10.3390/biomedicines11020617_

Round 1
Reviewer 1 Report
Manuscript Number: biomedicines-2174498
The manuscript of the review article prepared by Mickensone Andre, Madhavan Nair and Andrea D. Raymond “HIV Latency and Eradication Strategies in Nanomedicine” is supposed to give insight regarding the mechanisms of HIV latency and the latest nanomedical strategies to treat and eliminate HIV infection.
This is a very concise work, where the authors provide an overview of the topic which is of value for readers working in that field. I am not a strong expert in the biological part of the work, and my comments mainly will address information regarding nanomedical strategies. Some issues that need to be added and addressed before publication are given below.
1. Please make Figure 1 more easily readable.
2. I suggest that paragraph 4 is written in a very general way based on fragmentary resources. Introduction of this paragraph is not directly related to nanomedicines for HIV treatment. It is not clear why the authors decided to introduce with these nanosystems, it is necessary to underline which kind of nanoparticles have potential to be used for defined activities. If authors planned to introduce readers with classification of nanoparticles – unfortunately, the information provided herein is not sufficient.
3. Moreover, in the subparagraphs 4.1. and 4.2. only 2-3 literature sources were discussed. Please add more information and literature sources regarding these important topics.
4. Please justify the full information regarding reference nr.14 in the list.
5. Please add literature data from period 2021-2022 regarding topics included in the article.
6. Please use the same style points in the decryption of reference numbers, for example, line 93: [14][15], [16]; line 136: [30],[32], [33]; line 158: [45], [46],[47]–[49]; line 162: …proviruses [50]. versus line 164: … latency[51]; line 178: [57] [58]; line 286: … cells26,[88],[89],119.
Consequently, I do recommend accepting this manuscript for publication with major revision.
Reviewer 2 Report
The manuscript entitled “HIV Latency and Eradication Strategies in Nanomedicine” is suitable for publication in ‘Biomedicines’ after minor revision.
Comments
- Authors should expand nanomedicine part. This part must be more specific.
- Authors did not mention the pharmacokinetic properties of general HIV drugs. There must be a section. If there is a nanomedicine part, they must mention the pharmacokinetic properties. At this stage, the paper “Pharmaceuticals (Basel). 2022 Oct 12;15(10):1255. doi: 10.3390/ph15101255” will be beneficial.
- Authors should mention in figures, which drawing programs were used.
Round 2
Reviewer 1 Report
Accept in present form